# Modulation of ER Stress and Inflammation by S-Ketamine, R-Ketamine, and Their Metabolites in Human Microglial Cells: Insights into Novel Targets for Depression Therapy

**DOI:** 10.3390/cells14110831

**Published:** 2025-06-03

**Authors:** Marta Jóźwiak-Bębenista, Anna Wiktorowska-Owczarek, Małgorzata Siatkowska, Piotr Komorowski, Aneta Włodarczyk, Edward Kowalczyk, Paulina Sokołowska

**Affiliations:** 1Department of Pharmacology and Toxicology, Medical University of Lodz, Zeligowskiego 7/9, 90-752 Lodz, Poland; anna.wiktorowska-owczarek@umed.lodz.pl (A.W.-O.); edward.kowalczyk@umed.lodz.pl (E.K.); 2Laboratory of Molecular and Nanostructural Biophysics, Bionanopark, Dubois 114/116, 93-465 Lodz, Poland; m.siatkowska@bionanopark.pl (M.S.);; 3Division of Biophysics, Institute of Materials Science, Lodz University of Technology, Stefanowskiego 1/15, 90-924 Lodz, Poland; 4Department of Tumor Biology, Medical University of Lodz, Zeligowskiego 7/9, 90-752 Lodz, Poland; aneta.wlodarczyk@umed.lodz.pl

**Keywords:** antidepressants, CHOP, depression, ER stress, GRP78, inflammation, ketamine, ketamine metabolites, treatment-resistant depression, UPR pathway

## Abstract

Despite affecting millions worldwide, major depressive disorder (MDD) remains a therapeutic challenge, with approximately one-third of patients failing to respond to standard treatments. The need for innovative, molecularly driven therapies has turned attention to ketamine and its enantiomers. While S-ketamine is clinically approved for treatment-resistant depression (TRD), it has various psychoactive side effects and potential for abuse. Hence, it is necessary to identify alternative compounds, such as R-ketamine, and their metabolites (e.g., 2S,6S-hydroxynorketamine and 2R,6R-hydroxynorketamine, collectively referred to as HNKs). Emerging evidence suggests that the pathophysiology of MDD involves two processes regulated by the unfolded protein response (UPR): endoplasmic reticulum (ER) stress and neuroinflammation. As such, they represent promising therapeutic targets. The study provides the first direct comparison of ketamine enantiomers and their metabolites in modulating ER stress and inflammatory signaling in human microglial cells (HMC3), which play key roles in neuroimmune communication. Both S-ketamine and R-ketamine, along with their metabolites, significantly reduced both the expression and protein levels of CHOP and GRP78—two critical UPR components—under tunicamycin-induced ER stress conditions. Additionally, the compounds significantly decreased IL-6 levels and, to a lesser extent, IL-8 levels in lipopolysaccharide (LPS)-stimulated microglia, indicating anti-inflammatory potential. Taken together, these findings highlight a novel glia-targeted mechanism by which ketamine and its metabolites modulate ER stress and neuroinflammation. CHOP and GRP78 appear to be stress-responsive molecular markers within the UPR pathway. These results justify further in vivo validation and support the development of antidepressants with fewer psychoactive effects.

## 1. Introduction

Major depressive disorder (MDD) affects approximately 350 million people worldwide. Approximately 40% of individuals treated for a major depressive episode do not achieve remission after at least two first-line antidepressant treatment trials and are considered to have treatment-resistant depression (TRD). The poor efficacy of currently available antidepressants, which primarily act by modulating biogenic amine levels, is likely due to the complex pathophysiology of depression. Increasing evidence suggests that endoplasmic reticulum (ER) stress and inflammation are involved in the disorder. Elevated levels of pro-inflammatory cytokines have been observed in a proportion of depressed patients, particularly those refractory to standard treatment and with suicidal thoughts or attempts [1,2].

Evidence indicates that circulating immune cells and their released cytokines and chemokines can infiltrate the central nervous system (CNS), contributing to the development of neuroinflammation [3]. This process is typically characterized by the activation of microglia, which adopt a pro-inflammatory state and release large amounts of pro-inflammatory cytokines [4,5]. These, in turn, initiate changes in neuronal plasticity, affect brain tissue, modulate the monoaminergic system, and trigger neurotoxic processes in regions involved in emotion regulation, such as the hippocampus, amygdala, and prefrontal cortex [6]. Two pro-inflammatory cytokines that have been consistently linked to the pathophysiology of depression—particularly of TRD—are IL-6 and IL-8; their elevated levels have been associated with increased neuroinflammation, impaired neuroplasticity, and resistance to standard antidepressant therapies [7,8].

Inflammatory cytokines impact multiple intracellular structures in activated glial cells, including, e.g., ER, leading to structural and functional impairments. Dysregulation of the ER results in the accumulation of unfolded or misfolded proteins within its lumen, a condition known as ER stress. In response, the cells activate the unfolded protein response (UPR), a protective mechanism mediated by three ER-resident sensors: inositol-requiring enzyme 1 (IRE1), protein kinase R-like ER kinase (PERK), and activating transcription factor 6 (ATF6) [9,10]. Under normal conditions, these sensors are maintained in an inactive state through association with ER chaperones such as GRP78 (BiP) and GRP94. Upon ER stress, GRP78 dissociates from these sensors, initiating signaling cascades that can either support survival or trigger apoptosis, depending on the duration and intensity of the stress [11,12,13]. All three UPR branches converge on the activation of C/EBP homologous protein (CHOP), a transcription factor that promotes pro-apoptotic gene expression while repressing anti-apoptotic signals. CHOP, along with GRP78, is widely used as a molecular marker of ER stress severity and UPR activation. Moderate UPR activation in astrocytes and microglia may support neuroprotection via the release of neurotrophic factors; however, during chronic or unresolved ER stress, the response shifts toward a pro-apoptotic and pro-inflammatory state. This creates a vicious cycle where ER stress and inflammation amplify each other, contributing to neuronal dysfunction. Furthermore, elevated expression of UPR-related genes, such as *GRP78* and *CHOP,* has been reported in animal models and postmortem studies of patients with MDD, particularly those with TRD [14,15,16,17], highlighting the involvement of ER stress in depressive pathology.

A better understanding of the upregulation of UPR-related molecules in CNS cells, together with the elevated expression of inflammatory genes and cytokines, may provide new insights into the pathophysiology of depression, particularly TRD. Until recently, advances in antidepressant therapy had limited success in treating individuals with TRD [18,19]. This fact, together with low remission rates of depression, and the delayed onset of antidepressant therapies, i.e., longer than two weeks, indicate a need for new antidepressant drugs with novel mechanisms of action. Recent efforts in neuropsychiatric drug development suggest that some psychoactive substances may be repurposed as novel therapies. For example, ketamine is an effective and rapid-acting antidepressant (RAD); however, its use has various undesirable side effects such as dissociation, and the potential for abuse [20,21]. Ketamine exists as two enantiomers, viz. S-ketamine and R-ketamine, with R-ketamine having stronger and longer-lasting antidepressant effects than S-ketamine [21]. Despite this, only S-ketamine has been approved so far by the FDA for use as an antidepressant drug in TRD, as a nasal spray, although clinical trials of R-ketamine are currently underway [22,23,24]. Both enantiomers are metabolized extensively in the body via CYP2B6- and CYP3A4-mediated N-demethylation to S-norketamine and R-norketamine, respectively. Norketamine then undergoes further catabolism to hydroxynorketamines (HNKs) and dihydronorketamine [25,26]. Some ketamine metabolites are biologically active [27,28,29,30]. Notably, the 2S,6S-HNK and 2R,6R-HNK metabolites have been shown to exert antidepressant-like behavioral effects in rodent models [30,31,32], making them compounds of particular interest [33]. Understanding the mechanism of action of ketamine and its metabolites in the treatment of TRD could help in the development of safer and more effective therapies (Figure 1).

The precise mechanisms through which ketamine exerts its rapid and potent antidepressant effects are not yet fully understood. While modulation of the glutamatergic system—crucial for neuroplasticity—is considered a key factor [34,35], growing evidence also points to anti-inflammatory properties that may contribute to its fast onset of action [36,37]. Given the established link between neuroinflammation and TRD, there is increasing interest in compounds that exert both antidepressant and anti-inflammatory effects. One proposed mechanism involves modulation of the UPR pathway activated during ER stress. Such dual activity could be particularly relevant for patients with inflammation-associated treatment resistance and suicidal ideation. Therefore, the aim of this study was to determine the effects of S-ketamine, R-ketamine, and their metabolites on the main components of the UPR pathway following ER stress induction in human microglial cells. The study also examines the effects of the S- and R-ketamine isomers and their metabolites on the release of pro-inflammatory cytokines IL-6 and IL-8. It is hypothesized that S-ketamine, R-ketamine, and their HNK metabolites attenuate ER stress and reduce pro-inflammatory cytokine release in human microglial cells, thus targeting key mechanisms underlying TRD, associated with inflammation.

## 2. Materials and Methods

### 2.1. Reagents

The following reagents for microglial cell culture: Eagle’s Minimum Essential Medium (EMEM), Fetal Bovine Serum (FBS), Dulbecco’s Phosphate Buffered Saline (D-PBS) and Trypsin-EDTA solution were obtained from ATCC (Manassas, VA, USA). Tunicamycin, Lipopolysaccharide (LPS), Trypsin-EDTA solution and MTT (3-(4,5-dimethylthiazol-2-yl)-2,5-diphenyltetrazolium bromide) were obtained from Sigma-Aldrich (Saint Louis, MO, USA). The RNeasy Mini Kit was purchased from Qiagen (Germantown, WI, USA) and other reagents for Real-Time PCR, i.e., Custom PrimePCR™ Real-Time PCR Plates, iScript™ cDNA Synthesis Kit, 2xSsoAdvanced Universal SYBR Green Supermix, Prime PCR RT Control and Prime PCR Control Assay were purchased from Bio-Rad (Berkeley, CA, USA).

Human IL-6 DuoSet ELISA and Human IL-8 DuoSet ELISA kits were purchased from R&D Systems (Minneapolis, MN, USA). The Human HSPA5/Endoplasmic reticulum chaperone BiP ELISA Kit was obtained from EIAab Science (Wuhan, China). Reagents for Western blotting, i.e., Mini-Protean TGX Stain-Free Gels, 10× Tris/Glycine/SDS Buffer, Trans-Blot Turbo RTA Transfer Kit, Nitrocellulose, Trans-Blot Turbo 5× Transfer buffer, Clarity Western ECL Substrate Assay, 4× Laemmli Sample Buffer, 2-Mercaptoethanol, Precision Plus Protein All Blue Standards and Precision Plus Protein Unstained Standards were all purchased from Bio-Rad (Berkeley, CA, USA). Reagents for the lysis buffer (urea, thiourea, Tris, 3-[(3-Cholamidopropyl)dimethylammonio]-1-propanesulfonate hydrate (CHAPS), IPG Buffer pH 4–7, dithiotreitol (DTT), 2D Quant Kit and 2D Clean Up Kit were purchased from GE Healthcare (Little Chalfont, UK). Anti-DDIT3 and anti-mouse IgG (HRP) were purchased from Abcam (Cambridge, MA, USA).

### 2.2. Chemicals

S-ketamine hydrochloride and R-ketamine hydrochloride were purchased as enantiomerically pure compounds from Bio-Techne (R&D Systems, Abington, UK). S-ketamine (Cat. no. 4379/50) and R-ketamine (Cat. no. 6751/50) were each supplied at ≥98% enantiomeric purity, according to the manufacturer’s certificate of analysis. The compounds were used without further purification. Additionally, the metabolites 2R,6R-hydroxynorketamine hydrochloride (Cat. no. SML1873) and 2S,6S-hydroxynorketamine hydrochloride (Cat. no. SML1875) were purchased from Sigma-Aldrich (Merck, St. Louis, MO, USA) and used in accordance with standard laboratory protocols.

### 2.3. Cell Culture

The human microglial clone 3 cell line (HMC3) was obtained from ATCC (Manassas, VA, USA; Cat. No. CRL-3304) and cultured according to the manufacturer’s recommended protocol. Cells were maintained in EMEM supplemented with 10% FBS and 1% penicillin-streptomycin, at 37 °C in a humidified atmosphere with 5% CO_2_. Upon reaching approximately 80–90% confluency, cells were passaged using a 0.25% Trypsin-EDTA solution and reseeded at a density of 1–4 × 10^4^ cells/cm^2^ or adjusted as needed for subsequent experiments.

### 2.4. Real-Time Polymerase Chain Reaction (rtPCR)

Assessment of the expression of the genes associated with ER stress was performed as described previously [9]. Briefly, microglial cells were seeded onto culture flasks at a density of 1–1.5 × 10^6^ cells/flask and cultured for 24 h. Following this, tunicamycin (0.5 µg/mL), either alone or in combination with S-ketamine, R-ketamine, or their metabolites, was added to the culture and cells were further incubated for 24 h. The cells were then harvested, and the cell pellets resuspended in RLT lysis buffer for RNA isolation and purification using the RNeasy Mini Kit (Qiagen).

Reverse transcription was performed using the iScript™ cDNA Synthesis Kit in a CFX96 thermal cycler (Bio-Rad, Berkeley, CA, USA). The resulting cDNA and the 2xSsoAdvanced Universal SYBR Green Supermix reagent were added to appropriate wells on a 96-well Custom PrimePCR™ Real-Time PCR Plate according to the manufacturer’s protocol (Bio-Rad, Berkeley, CA, USA). Real-time PCR was then conducted in a CFX96 thermal cycler (Bio-Rad, Berkeley, CA, USA), and data analysis was performed using the 2^−ΔΔCq^ method. GAPDH and TBP were chosen as reference genes for normalization. The results were analyzed using one-way ANOVA with a significance level of *p* < 0.05. A fold-change value above 1.5 was considered significant.

### 2.5. Cell Viability

Cell viability was assessed using the colorimetric MTT assay. Cells were seeded onto 96-well plates at a final density of 7 × 10^3^ cells per well and cultured for 24 h. They were then treated with tunicamycin (0.5 μg/mL) or LPS (10 ng/mL), S-ketamine, R-ketamine, or their metabolites (0.1–100 µM), and incubated for 24 or 48 h. After incubation, MTT solution was added to the cell culture for another four hours. Absorbance was measured at 570 nm using an EL ×800 microplate reader (BioTek, Winooski, VT, USA); the final value was directly proportional to the number of viable cells. Cell viability was calculated using the formula: viability [%] = (A/AC) × 100%, where A represents the absorbance of the investigated sample and AC represents the absorbance of the control (untreated cells).

### 2.6. ELISA Tests

The release of the GRP78 protein was measured with the Human HSPA5/Endoplasmic reticulum chaperone BiP ELISA Kit. IL-6 and IL-8 levels were measured using the Human IL-6 DuoSet ELISA or Human IL-8 DuoSet ELISA kits. Microglia were seeded onto 12-well plates at a density of 1.5 × 10^5^ cells per well. After 24 h, the cells were exposed to tunicamycin (0.5 μg/mL) or LPS (10 ng/mL) alone for 24 or 48 h, or in combination with S-ketamine/R-ketamine or their metabolites. The cell supernatants were then collected, centrifuged at 250× *g* for 5 min to remove cellular debris, subjected to analysis according to the manufacturer’s protocol, and measured using an ELx800 microplate reader (BioTek, Winooski, VT, USA).

### 2.7. Protein Extraction and Western Blot Analysis

CHOP protein expression was analyzed using Western blotting. The cells were seeded onto culture flasks at a final density of 2.5–3 × 10^6^ cells. After 24 h of culture, the cells were treated with tunicamycin (0.5 µg/mL) or tunicamycin in combination with S-ketamine/R-ketamine or their metabolites (10 µM) for 48 h. Following treatment, the cells were washed with PBS, trypsinized, and centrifuged at 150× *g* for 5 min at room temperature. Cell pellets were resuspended in lysis buffer containing 7 M urea, 2 M thiourea, 4% *w*/*v* CHAPS, 2% *v*/*v* IPG buffer, and 40 mM DTT. Protein concentrations were determined using the 2D Quant Kit according to the manufacturer’s protocol, and any contaminants were removed using the 2D Clean Up Kit.

Equal amounts of protein extracts (25 μg per lane) were separated on an 8% SDS-polyacrylamide gel and transferred onto a PVDF membrane. The membranes were blocked with 3% (*w*/*v*) non-fat dry milk prepared in TBST (Tris-buffered saline with 0.1% Tween-20) for one hour at room temperature. Subsequently, the membranes were incubated overnight at 4 °C with primary antibodies against CHOP (Anti-DDIT3/CHOP (L63f7), Mouse mAb, #2895, Cell Signaling Technology, Danvers, MA, USA, 1:500) and beta-tubulin (anti-beta-tubulin, Abcam, Cambridge, UK, #ab131205, 1:10,000).

After incubation, membranes were washed three times with TBST and incubated with an HRP-linked secondary antibody (anti-mouse IgG, #7076, Cell Signaling Technology, 1:2500) for one hour at room temperature, followed by additional washes in TBST. Chemiluminescence detection was performed using the Clarity Max Western ECL Substrate kit (Bio-Rad), and signals were visualized with the ChemiDoc MP Imaging System (Bio-Rad). Densitometric analysis was conducted using ImageJ software (version 1.53t, National Institutes of Health, Bethesda, MD, USA), and CHOP expression was normalized to beta-tubulin bands.

### 2.8. Data Analysis

Data are expressed as mean ± standard error of the mean (SEM). Normality was assessed using the Shapiro–Wilk test, and homogeneity of variances using Levene’s test. When these assumptions were met, data were analyzed using one-way ANOVA followed by Tukey’s *post hoc* test or unpaired *t*-test, as appropriate. All tests were two-tailed, and statistical significance was set at *p* < 0.05. Statistical analyses were performed using GraphPad InStat version 9.3.0 (GraphPad Software, San Diego, CA, USA).

## 3. Results

### 3.1. Effect of S-Ketamine (S-Ket), R-Ketamine (R-Ket), and Their Metabolites on the Viability of the Microglial Cell Line

The concentrations of drugs and substances inducing ER stress and inflammation were selected based on previous experiments and literature data [9,38,39]. To assess their potential cytotoxicity, all studied compounds (S-ketamine, R-ketamine, and their metabolites) were applied at the full range of concentrations (0.1, 1, 10, and 100 µM). It was found that 10 µM is a non-toxic dose (Appendix A). While a direct extrapolation to in vivo conditions is limited, this dose is pharmacologically relevant, considering that clinical ketamine administration (e.g., 0.5 mg/kg IV) leads to plasma concentrations of approximately 1–4 µM [29]. When applied at the tested concentration (10 µM), neither R-ketamine, S-ketamine, or their metabolites affected the viability of microglial cells after 24 and 48 h of incubation, and neither did LPS at 10 ng/mL, similar to the tested drugs. However, tunicamycin at a concentration of 0.5 µg/mL reduced cell viability to 92% after 24 h and to 76% after 48 h, suggesting a cytotoxic effect over time (moderate ER stress) (Table 1).

### 3.2. Effect of S-Ketamine (S-Ket), R-Ketamine (R-Ket), S-Ketamine Metabolite (2S,6S-HNK) and R-Ketamine Metabolite (2R,6R-HNK) on the Expression of UPR Pathway Genes

To investigate the impact of S-ketamine, R-ketamine, and their metabolites on the expression of UPR pathway genes in human microglial cells, the most characteristic genes from the three branches of the UPR pathway were selected for the study. Following a 24 h incubation with 0.5 µg/mL tunicamycin (a known ER stress inducer), several key genes in the UPR pathway exhibited significantly increased expression (F(6, 17) = 44.70, *p* < 0.0001). Specifically, *DDIT3* and *HSPA5* were markedly upregulated, with fold changes of 12.22 and 10.14, respectively. Additionally, *ATF4* and *ATF6* gene expression increased by 2.33-fold and 2-fold, respectively. Other genes such as *CREB3L1*, *EDEM1*, and *ERN1* showed moderate increases in expression, with respective fold change values of 1.17, 1.83, and 1.41. In the absence of stress, neither S-ketamine nor R-ketamine nor their metabolites significantly affected the expression of the UPR-related genes after 24 h of incubation when administered at 10 μM (Table 2).

However, under tunicamycin-induced ER stress, S-ketamine, R-ketamine, the S-ketamine metabolite (2S,6S-HNK) and the R-ketamine metabolite (2R,6R-HNK) significantly reduced the expression of the *DDIT3* gene (F = 14.50–43.18, *p* < 0.0001) (Figure 2). The strongest effect was observed with S-ketamine and its metabolite (2S,6S-HNK), which induced significant fold changes in *DDIT3* gene expression of 7.51 and 6.25, respectively. The difference in *DDIT3* expression fold change (Δ) between tunicamycin alone and co-treatment with S-ketamine or its metabolite (2S,6S-HNK) was 4.71 and 5.97, respectively. These results are consistent with the literature data suggesting that ketamine’s therapeutic effects are mediated through its active metabolites.

In comparison, R-ketamine yielded an 8.06-fold change in *DDIT3* expression, and its metabolite an 8.86-fold change; these correspond to respective differences of 4.16 and 3.26-fold compared to tunicamycin alone. Additionally, under tunicamycin-induced ER stress, both S-ketamine and its metabolite reduced the expression of the *HSPA5* gene, which also plays an important part in the ER stress response. However, this reduction was much weaker and not statistically significant.

### 3.3. Effect of S-Ketamine (S-Ket), R-Ketamine (R-Ket), and Their Metabolites on GRP78 Protein Release Under ER Stress Conditions

Since differences in *HSPA5* gene expression in human microglial cells were observed after 24 h of incubation with tunicamycin, the incubation period was extended to 48 and 72 h to account for potential protein synthesis (Figure 3A). Tunicamycin (0.5 µg/mL) significantly increased the GRP78 protein level compared to the control, and this effect was more pronounced with longer incubation times. Specifically, the level of GRP78 was 23% above control values after 24 h, rising to 120% after 48 h and to 143% after 72 h, confirming a robust ER stress response to tunicamycin. The 24 and 48 h incubations with 0.5 µg/mL tunicamycin were selected for further studies as an in vitro model of ER stress.

Incubation with S-ketamine, R-ketamine, 2S,6S-HNK or 2R,6R-HNK alone did not affect the level of GRP78, either for 24 h or for 48 h incubation. In addition, no significant change in GRP78 protein release was observed under 24 h tunicamycin-induced ER stress (Figure 3B). However, a significant reduction in GRP78 level was noted for all compounds after 48 h of incubation with tunicamycin (Figure 3C; F(5, 15) = 12.53, *p* < 0.001). The strongest effect was observed for 2R,6R-HNK, which significantly lowered GRP78 level by 36%, followed by R-ketamine, which reduced it by 26%. A similar effect was noted for 2S,6S-HNK. The weakest effect was observed for S-ketamine, which reduced GRP78 level by 21%; this reduction was not statistically significant (Figure 3C).

### 3.4. Effect of S-Ketamine (S-Ket), R-Ketamine (R-Ket), and Their Metabolites on the CHOP Protein Expression Under ER Stress Condition

Due to the lack of consistent results in the ELISA assay for CHOP, its protein expression was analyzed by Western blot; such confirmatory assessment, i.e., using both ELISA and Western blot, is often necessary for proteins such as CHOP with low expression or inconsistent immunoassay performance.

The significant increase in DDIT3 gene expression observed after 24 h of incubation with tunicamycin translated into elevated CHOP protein levels after 48 h. Treatment with tunicamycin for 48 h significantly increased CHOP protein expression compared to control cells (F(5, 12) = 12.12, *p* = 0.0002). Western blot analysis confirmed that co-treatment with S-ketamine and R-ketamine effectively reduced CHOP protein expression. Furthermore, the ketamine metabolites (2R,6R-HNK and 2S,6S-HNK) demonstrated a comparable effect to the parent compounds, similarly decreasing CHOP protein levels (Figure 4).

### 3.5. Effect of S-Ketamine (S-Ket), R-Ketamine (R-Ket), and Their Metabolites on IL-6 and IL-8 Release Under Inflammatory Conditions

The next stage examined the effect of S-ketamine (S-ket), R-ketamine (R-ket), and their metabolites on IL-6 and IL-8 release in an in vitro neuroinflammation model of LPS-treated microglial cells (Figure 5). It was found that 24 h and 48 h incubation with LPS treatment significantly increased IL-6 and IL-8 levels in microglial cells compared to untreated controls (*p* < 0.001). The drugs alone did not affect IL-6 and IL-8 release. One-way ANOVA revealed a significant effect of treatment on IL-6 levels at 24 h (F(5, 25) = 10.66, *p* < 0.0001) and 48 h (F(5, 17) = 12.03, *p* < 0.0001), as well as on IL-8 levels at 24 h (F(5, 28) = 26.02, *p* < 0.0001) and 48 h (F(5, 23) = 4.88, *p* = 0.0034).

S-ketamine, R-ketamine, 2S,6S-HNK, and 2R,6R-HNK significantly reduced IL-6 levels in microglial cells following 24 and 48 h LPS stimulation, by approximately 50%, compared to LPS-treated controls (Figure 5A). The effect was comparable between the parent compounds and their metabolites and remained stable across both time points, indicating no time-dependent variation.

Significantly weaker effects were observed on IL-8, which also has pro-inflammatory properties (Figure 5B). After 24 h treatment, the tested drugs reduced IL-8 levels by a mean value of 28% compared to LPS-treated controls. The strongest effect was observed for S-ketamine, which reduced IL-8 levels by 33%, with the other compounds demonstrating weaker and insignificant reductions. After 48 h incubation, only S-ketamine and, to a lesser extent, R-ketamine reduced IL-8 level further, but these effects were not statistically significant.

### 3.6. Effects of S-Ketamine (S-Ket), R-Ketamine (R-Ket), and Their Metabolites on IL-6 and IL-8 Release Under ER-Stress Conditions

Tunicamycin significantly elevated IL-6 level and reduced IL-8 release after 24 and 48 h of incubation in microglial cells. For IL-6, the tested drugs reduced the levels relative to cells treated with tunicamycin alone. The result was significant at 48 h (F(5, 25) = 3.05, *p* = 0.0279; one-way ANOVA), although no differences between individual groups were noted in the post hoc analysis, and no significant effect was found at 24 h (F(5, 30) = 2.17, *p* = 0.0841). For IL-8, no significant changes in IL-8 release were noted for any treatment at either time point compared to tunicamycin alone (24 h: F(5, 29) = 1.84, *p* = 0.1363; 48 h: F(5, 32) = 2.23, *p* = 0.0749) (Figure 6).

## 4. Discussion

Although the clinical efficacy of ketamine in TRD is well established, the precise molecular mechanisms underlying its rapid antidepressant action remain poorly understood. Most research has focused on synaptic mechanisms, particularly glutamatergic transmission and neuroplasticity; however, recent studies suggest that inter alia ER stress and UPR signaling may also be involved. Furthermore, little is known about how ketamine and its metabolites affect these signaling mechanisms in glial cells, particularly microglia, which play key roles in neuroinflammation.

Animal studies have shown that ketamine can alter the expression of UPR-related proteins, including *PERK*, *IRE1*, and *CHOP*, and that these effects may differ depending on the region of the brain [40]. Additionally, evidence suggests that the UPR pathway may influence the mTOR pathway, both of which are critical for maintaining cellular homeostasis and survival; also, mTOR may modulate the antidepressant effects of ketamine. Additionally, ketamine has been found to increase the expression of ER stress-related proteins in adult neural stem cells, although only at high concentrations (i.e., 400 µM) far exceeding those used clinically [41]. As S-ketamine, and to a lesser extent R-ketamine, have been found to significantly activate key genes of the classical UPR pathway in astrocytes under ER stress [9], one goal of the present study was to determine how these drugs may influence other glial cells.

The choice of microglia for this study was not accidental. Microglial cells play a central role in maintaining homeostasis in the CNS: they regulate inflammation by releasing inflammatory cytokines, clear apoptotic cells via phagocytosis, and maintain synaptic plasticity and neural network formation through synapse pruning. Microglia generally serve a protective function; however, under pathological conditions, such as depression or neurodegenerative diseases, these cells become overactivated, leading to the release of pro-inflammatory cytokines, such as IL-6 and IL-8, and increased neuroinflammation [7,42]. Given their unique properties as the primary immune effector cells in the brain, microglial cells are an ideal model for studying neuroinflammatory mechanisms. Their ability to respond to stimuli such as LPS or ER stress makes them not only useful markers of inflammation, but also versatile models for studying neuroimmune mechanisms in neuropsychiatric disorders. Microglia have played an essential part in many in vitro and in vivo studies examining the immunomodulatory action of ketamine [43]. In this study, they were used to determine how ketamine and its metabolites impact key pathological mechanisms associated with neuroinflammation and ER stress, particularly in the context of depression and TRD.

The present study provides new insights into how S-ketamine, R-ketamine, and their metabolites modulate ER stress responses and inflammatory signaling in human microglial cells, two processes known to be involved in neuroinflammation and depression. To evaluate the potential contribution of the UPR, the study examined the effects of 24 h exposure to tunicamycin, an ER stress inducer, in the microglial cells. It was found that tunicamycin stimulation significantly upregulated the levels of *GRP78*, *CHOP*, *ATF6* and *ATF4*, a downstream target of the PERK pathway. The IRE1 pathway was also activated, but to a lesser degree. In addition, under ER stress conditions (24 h tunicamycin treatment), all four treatments significantly reduced the expression of *CHOP*, with a more pronounced reduction observed for S-ketamine and its metabolite (2S,6S-HNK) compared to R-ketamine and its metabolite (2R,6R-HNK). Additionally, S-ketamine and its metabolite also reduced GRP78 expression.

The next stage examined whether the observed changes in gene expression translates into the inhibition of protein synthesis, such as CHOP (encoded by *DDIT3*) and GRP78 (encoded by *HSPA5*). All tested compounds reduced CHOP protein levels following 48 h of incubation with tunicamycin; this suggests that treatment may help attenuate the ER stress response and reduce pro-apoptotic signaling associated with CHOP activation. Although R-ketamine and its metabolite did not significantly affect GRP78 gene expression, both compounds reduced GRP78 protein levels. Notably, the R-ketamine metabolite (2R,6R-HNK) exerted a stronger effect than the other three treatments. This effect was significant after the longer incubation (48 h) with tunicamycin.

S-ketamine, R-ketamine, and their metabolites were found to reduce the levels of two key proteins of the UPR pathway in microglial cells, GRP78 and CHOP, which are markers of ER stress activation. Interestingly, although the GRP78 protein levels were reduced following ketamine treatment, no significant changes in the corresponding gene (*HSPA5*) were found at the mRNA level. This difference may be attributed to post-transcriptional regulation, such as altered translation efficiency or changes in protein turnover. It is not uncommon for mRNA and protein levels to be uncoupled, especially in the case of molecular chaperones like GRP78, whose activity is tightly regulated under stress conditions.

The crosstalk between ER stress and inflammation in the brain has been widely described [40,44,45]. ER stress contributes to inflammation through activation of the CHOP-mediated UPR pathway; this is believed to drive the production of pro-inflammatory cytokines and reactive oxygen species (ROS), which may further exacerbate neuroinflammation associated with depression and TRD. Notably, a study conducted on BV-2 microglial cells demonstrated that suppression of *CHOP* expression using small interfering RNA (siRNA) targeting the Ddit3 gene reduced IL-6 levels by impairing the JAK/STAT signaling pathway. This suggests that CHOP may have central role in sustaining inflammation, being a critical regulator of IL-6 expression in response to proteasome impairment. Hence CHOP may represent a potential therapeutic target in conditions associated with neuroinflammatory processes, such as depression [46]. Similarly, our present findings also indicate that the tested ketamine isomers and their metabolites inhibited the increase in UPR pathway proteins following LPS treatment, manifested as a marked reduction in IL-6 levels and a weaker effect on IL-8. This lesser effect on IL-8 secretion may reflect fundamental differences in their regulation. While IL-6 expression is closely linked to CHOP-mediated ER stress signaling, IL-8 production is largely controlled by pathways such as NF-κB and AP-1, which may be less responsive to modulation via the UPR. Indeed, previous studies have shown that IL-8 transcription depends more on TLR4-NF-κB activation and is less sensitive to ER stress inhibitors [47,48]. Notably, our findings indicate that of the tested drugs, S-ketamine exhibited the strongest inhibitory effect on both IL-6 and IL-8 levels. Importantly, all compounds were tested at a concentration of 10 μM, which did not significantly affect cell viability, as confirmed by the MTT reduction assay (Table 1); as such, the reduced inflammatory mediator levels were not associated with any cytotoxic effects. These findings confirm that ketamine isomers and their metabolites play a potential role in modulating neuroinflammatory responses.

Over the past decade, numerous studies have explored the immunomodulatory properties of ketamine in MDD, and attribute its rapid antidepressant effects to its ability to regulate inflammation. Ketamine has been frequently observed to modulate the levels of IL-1β, IL-6, and TNF-α, all of which are strongly implicated in depressive illness [49]. For instance, higher baseline levels of IL-6 and IL-1β, and their reduction following ketamine treatment, were associated with better treatment outcomes; however, the baseline cytokine levels in their study were notably higher than those typically observed in most populations [50]. Similarly, IL-8 has emerged as a potential biomarker for predicting response to ketamine treatment. Studies in male patients have shown that a decrease in IL-8 levels after ketamine administration correlates with improved outcomes in TRD [51].

Ketamine has also demonstrated anti-inflammatory potential in vitro, where it was observed to inhibit the production and release of pro-inflammatory cytokines, such as IL-6, IL-1β and TNF-α, in murine macrophages [52], primary rat microglial cells [53] and human-derived glioblastoma cells [54]. Moreover, ketamine was also shown to reduce the expression of pro-inflammatory cytokines and enzymes in an LPS-induced inflammation model based on BV-2 microglial cells [55].

Several investigators have also explored whether HNKs exert anti-inflammatory effects. For instance, a recent study by Ho et al. used cultures of a human microglial cell line (HMC3 cells) and performed transcriptome analysis following a 24 h exposure to 2S,6S-HNK and 2R,6R-HNK (400 nM) [33,56]. The authors reported a significant upregulation of type I interferon pathway activity induced by both HNKs. They also observed increased expression and nuclear translocation of signal transducer and activator of transcription 3 (STAT3), a key transcription factor regulating interferon pathways and gene expression. In the present study, HNKs also demonstrated anti-inflammatory effects in microglial cells, lowering IL-6 levels to a similar degree as the parent compounds, S-ketamine and R-ketamine, following LPS stimulation. This effect was observed for both 2S,6S-HNK and 2R,6R-HNK at 24 and 48 h, indicating sustained anti-inflammatory activity over time. Hence, it appears that HNKs may retain the anti-inflammatory properties of their parent compounds and modulate microglial responses to inflammatory stimuli. However, the potential anti-inflammatory effects of HNKs require further investigation, including an assessment of their broader impact on cytokines and other inflammatory pathways.

Contrary to our expectations, the ketamine isomers and their metabolites did not significantly affect the tunicamycin-induced release of IL-6 and IL-8, suggesting that they may not inhibit the ER stress-associated inflammatory response under these experimental conditions. The fact that the treatment modulated the UPR pathway without a corresponding change in cytokine release suggests that its anti-inflammatory effects might be independent of ER stress; alternatively, they could be mediated through alternative pathways not examined in this study. It is important to note that while tunicamycin is a well-characterized inducer of ER stress, it may not fully represent all the mechanisms of ER stress relevant to neuroinflammation and depression. Interestingly, our findings indicate that tunicamycin-induced ER stress affects interleukin levels differently to LPS or TNF-α induction, as it increases IL-6 but decreases IL-8; this difference remains unexplained and warrants further investigation [57]. Moreover, tunicamycin had a much stronger impact on interleukin levels than LPS, potentially due to its cytotoxic effects over time when applied at 0.5 µg/mL. In contrast, LPS at a concentration of 10 ng/mL did not induce noticeable cytotoxicity and elicited a milder increase in interleukin levels.

The weak effect of the ketamine isomers and their metabolites on IL-6 and IL-8 levels in the present experimental setup may be attributed to the fact that moderate ER stress was induced by tunicamycin. It is possible that the tested compounds act earlier, during the activation phase of the UPR pathway or under conditions of mild ER stress, where their modulatory effects might be more pronounced [57]. It remains uncertain whether our findings, based on tunicamycin induction, can be extrapolated to other ER stress inducers, such as thapsigargin, or to pathophysiological conditions in vivo.

A limitation of our study is the exclusive use of the HMC3 microglial cell line, which may not fully reflect the phenotype and functionality of primary microglia in vivo. Therefore, further validation using primary cultures or animal models is warranted to widen the generalizability of these findings.

## 5. Conclusions

This study provides new insights into how ketamine isomers (S-ketamine and R-ketamine) and their metabolites (2S,6S-HNK and 2R,6R-HNK) modulate ER stress and neuroinflammation, two processes increasingly recognized as key contributors to the pathophysiology of MDD and TRD. All tested substances were found to downregulate markers of the UPR pathway and lower pro-inflammatory cytokine levels in microglial cells. As such, our findings suggest that these compounds may influence neuroimmune signaling. Interestingly, the metabolites mimicked the effects of the parent compounds, which may justify further preclinical exploration, particularly considering their possibly lower psychoactive liability. Our in vitro results indicate that CHOP and GRP78 are stress-responsive molecular markers involved in the mechanism of action of ketamine isomers and their metabolites: representative RADs. These findings provide a rationale for future in vivo studies to assess their translational relevance in TRD.

## Figures and Tables

**Figure 1 cells-14-00831-f001:**
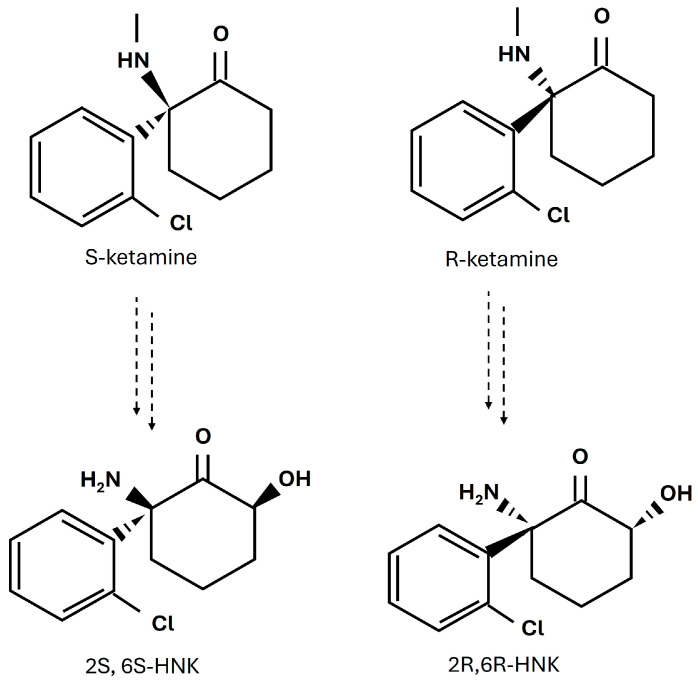
Chemical structures of ketamine enantiomers, S-ketamine and R-ketamine, and their major metabolites, 2S,6S-HNK and 2R,6R-HNK.

**Figure 2 cells-14-00831-f002:**
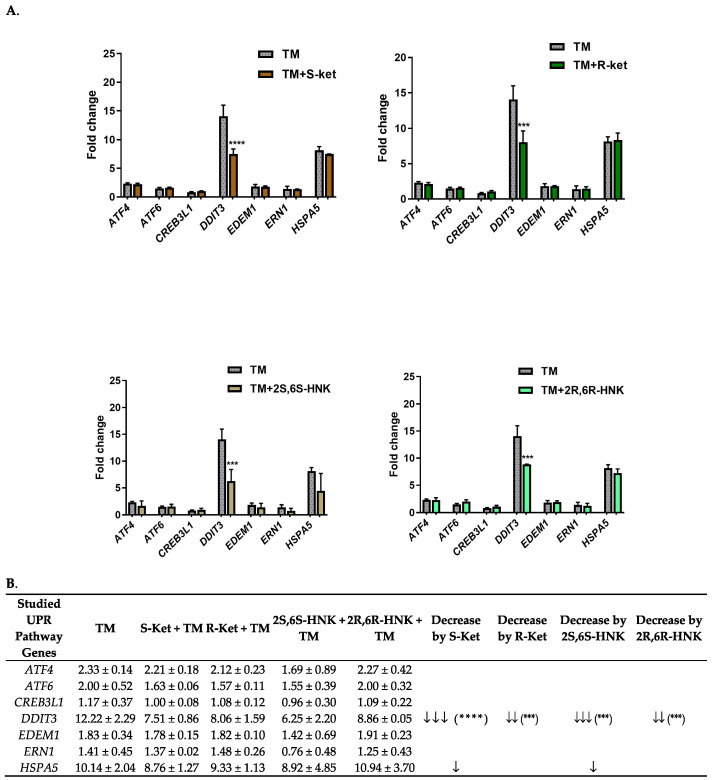
(**A**) Effect of S-ketamine (S-ket, 10 µM), R-ketamine (R-ket, 10 µM), S-ketamine metabolite (2S,6S-HNK, 10 µM) and R-ketamine metabolite (2R,6R-HNK, 10 µM) on the expression of UPR pathway genes exposed to tunicamycin (TM, 0.5 µg/mL) for 24 h in microglial cells. (**B**) To illustrate differences in gene expression between cells treated with drug under ER stress conditions and those treated with tunicamycin alone, arrows indicate the extent of downregulation: ↓ for 1–2 fold change; ↓↓ 3–4.5 fold change; ↓↓↓ > 4.5 fold change. Data are presented as the mean ± SEM (*n* = 4–6) and expressed as fold change vs. untreated control cells. Statistical significance vs. tunicamycin-treated cells is indicated as follows: **** *p* < 0.0001; *** *p* < 0.001.

**Figure 3 cells-14-00831-f003:**
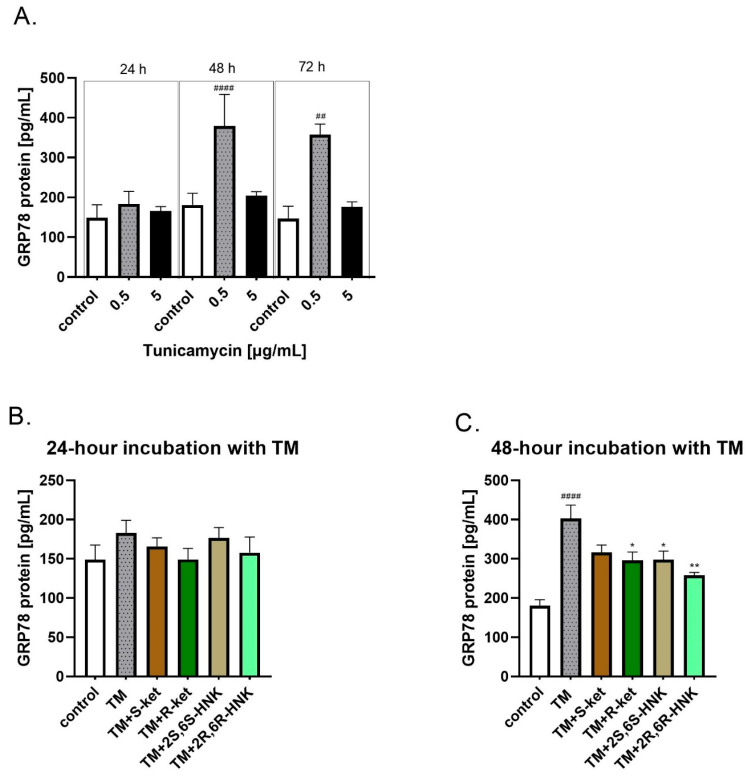
(**A**) Levels of GRP78 protein in human microglial cells after treatment with various concentrations of tunicamycin (0.5, and 5 µg/mL) over different time periods (24, 48, and 72 h); (**B**,**C**) effect of R-ketamine (R-ket, 10 µM), S-ketamine (S-ket, 10 µM), R-ketamine metabolite (2R,6R-HNK, 10 µM) and S-ketamine metabolite (2S,6S-HNK, 10 µM) on GRP78 release from microglial cells exposed to 24 h and 48 h tunicamycin (TM, 0.5 µg/mL). Data are presented as the mean ± SEM (*n* = 4–6) and expressed as a concentration of respective protein. Statistical significance tunicamycin vs. control ^####^ *p* < 0.0001, ^##^ *p* < 0.01; TM-treated cells vs. drug-treated cells under TM-induced ER stress: * *p* < 0.05; ** *p* < 0.01.

**Figure 4 cells-14-00831-f004:**
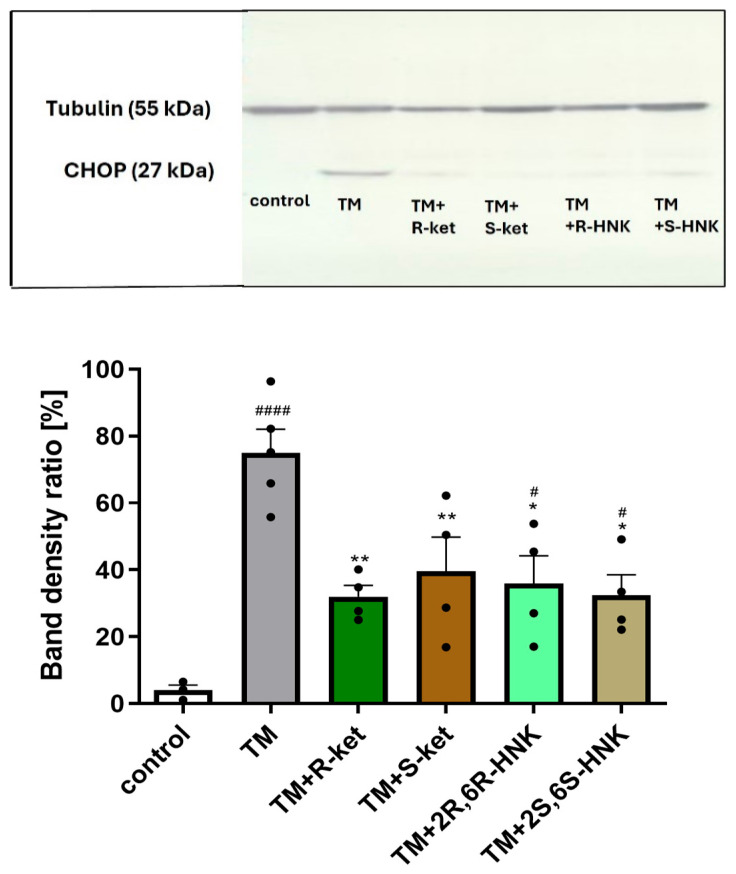
Effects of tunicamycin (TM, 0.5 µg/mL) alone, and R-ketamine (R-ket, 10 µM), S-ketamine (S-ket, 10 µM), R-ketamine metabolite (2R,6R-HNK, 10 µM), and S-ketamine metabolite (2S,6S-HNK, 10 µM) on CHOP protein expression in microglial cells after 48 h of tunicamycin-induced ER stress. The top panel shows representative Western blot bands of CHOP and tubulin (used for normalization). The bottom panel presents the mean data of densitometric analysis of CHOP expression. Data are presented as mean ± SEM (*n* = 4–6). Statistical significance: tunicamycin vs. control, ^####^ *p* < 0.0001, ^#^ *p* < 0.05; TM-treated cells vs. drug-treated cells under TM-induced ER stress, ** *p* < 0.01, * *p* < 0.05.

**Figure 5 cells-14-00831-f005:**
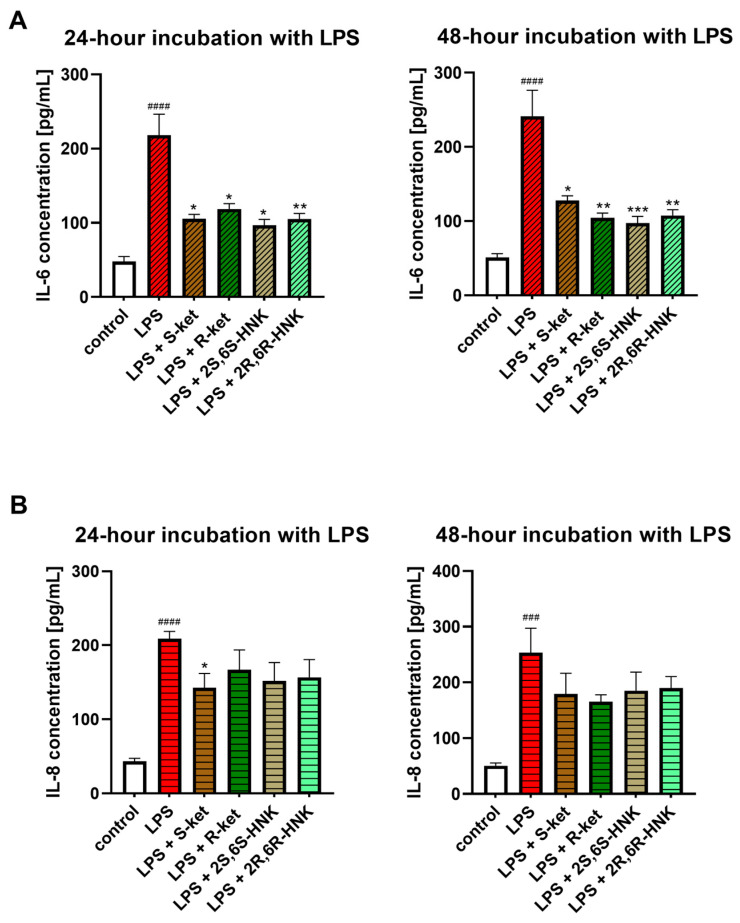
Effect of R-ketamine (R-ket, 10 µM), S-ketamine (S-ket, 10 µM), R-ketamine metabolite (2R,6R-HNK, 10 µM) and S-ketamine metabolite (2S,6S-HNK, 10 µM) on IL-6 release (**A**) or IL-8 secretion (**B**) from microglial cells exposed to 24 h and 48 h lipopolysaccharide (LPS, 10 ng/mL). Data are presented as the mean ± SEM (*n* = 6–8) and expressed as a concentration of respective protein. Statistical significance: LPS vs. control ^###^ *p* < 0.001; ^####^ *p* < 0.0001; LPS-treated cells vs. drug-treated cells under LPS-induced inflammation: * *p* < 0.05; ** *p* < 0.01; *** *p* < 0.001.

**Figure 6 cells-14-00831-f006:**
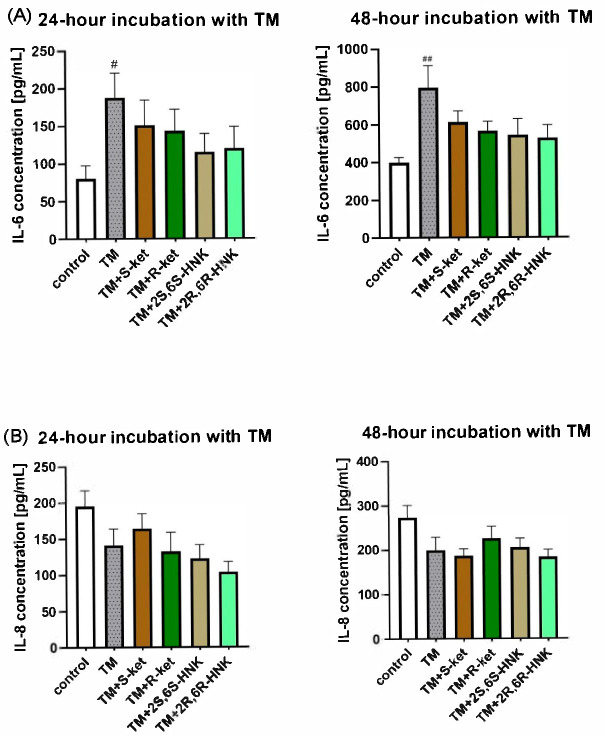
Effect of R-ketamine (R-ket, 10 µM), S-ketamine (S-ket, 10 µM), R-ketamine metabolite (2R,6R-HNK, 10 µM) and S-ketamine metabolite (2S,6S-HNK, 10 µM) on IL-6 release (**A**) or IL-8 secretion (**B**) from microglial cells exposed to 24 h and 48 h tunicamycin (TM, 0.5 µg/mL). Data are presented as the mean ± SEM (*n* = 6–8) and expressed as a concentration of respective protein. Statistical significance: TM vs. control ^##^ *p* < 0.01; ^#^ *p* < 0.01.

**Table 1 cells-14-00831-t001:** The effect of S-ketamine (S-ket, 10 µM), R-ketamine (R-ket, 10 µM), S-ketamine metabolite (2S,6S-HNK, 10 µM), R-ketamine metabolite (2R,6R-HNK, 10 µM), tunicamycin (TM, 0.5 µg/mL) and lipopolysaccharide (LPS, 10 ng/mL) on the viability of microglial cells following 24 h and 48 h incubation. Data are presented as the mean ± SEM (*n* = 8) and expressed as a percentage of untreated control cells.

		% Control ± SEM			
Time of Incubation		24-h		48-h	
control		100	±1.96	100	±0.51
S-ket	10 µM	108	±3.83	99	±0.66
R-ket		115	±8.43	102	±0.83
2S,6S-HNK		103	±3.89	102	±0.8
2R,6R-HNK		100	±3.52	98	±1.33
TM	0.5 µg/mL	92	±5.48	76	±2.26
LPS	10 ng/mL	100	±2.59	106	±2.42

**Table 2 cells-14-00831-t002:** Effects of S-ketamine (S-ket, 10 µM), R-ketamine (R-ket, 10 µM), an S-ketamine metabolite (2S,6S-HNK, 10 µM) and an R-ketamine metabolite (2R,6R-HNK, 10 µM) on the expression of UPR pathway genes in microglial cells after 24 h of incubation. Data are presented as mean ± SEM (*n* = 3–4) and expressed as fold change vs. untreated control cells. * According to https://www.proteinatlas.org/ (accessed on 20 May 2025); ^#^ more frequent in usage in the literature.

Studied UPR Pathway Genes	Alternative Name	Encoded Protein *	Fold Change			
			S-Ket	R-Ket	2S,6S-HNK	2R,6R-HNK
* ATF4 *	* CREB-2 *	* Activating transcription factor 4 *	1.23 ± 0.25	1.10 ± 0.13	1.00 ± 0.17	1.09 ± 0.22
* ATF6 *	* - *	* Activating transcription factor 6 *	1.07 ± 0.08	1.31 ± 0.20	1.53 ± 0.53	1.09 ± 0.09
* CREB3L1 *	* Oasis *	* CAMP responsive element binding protein 3 like 1 *	0.99 ± 0.29	1.01 ± 0.22	1.04 ± 0.34	0.89 ± 0.29
* DDIT3 *	* CHOP ^#^ *	*DNA damage inducible transcript 3/C/EBP-homologous protein*	1.09 ± 0.14	0.98 ± 0.04	0.93 ± 0.10	1.07 ± 0.11
* EDEM1 *	* EDEM *	* ER degradation enhancing alpha-mannosidase like protein 1 *	0.99 ± 0.19	1.03 ± 0.12	1.10 ± 0.13	0.85 ± 0.23
* ERN1 *	* IRE1 *	* Endoplasmic reticulum to nucleus signalling 1/Inositol-requiring enzyme 1 *	0.84 ± 0.15	0.83 ± 0.10	0.95 ± 0.24	0.69 ± 0.21
* HSPA5 *	* GRP78 *	* Heat shock protein family A (Hsp70) member 5 *	1.19 ± 0.32	1.35 ± 0.40	1.08 ± 0.35	1.15 ± 0.26

## Data Availability

All data are contained within the article.

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
