# Peer review of "Modulation of ER Stress and Inflammation by S-Ketamine, R-Ketamine, and Their Metabolites in Human Microglial Cells: Insights into Novel Targets for Depression Therapy"

_cells, 2025, doi:10.3390/cells14110831_

Round 1

Reviewer 1 Report (Previous Reviewer 2)

Comments and Suggestions for Authors

 The manuscript has been  improved so I consider it acceptable in present form

Author Response

Thank you for your positive evaluation.

Reviewer 2 Report (Previous Reviewer 1)

Comments and Suggestions for Authors

Dear Marta Jozwiak-Bebnista and co-authors,

Your study "Modulation of ER stress and inflammation by (S)-ketamine, (R)-ketamine, and their metabolites in human microglial cells: Insights into novel targets for depression therapy" build up the strong basis for in vivo experiments in near future. The revised version of the manuscript is fully satisfactory and I don't have any concerns. 

Author Response

Thank you for your kind assessment. We are glad the manuscript is now acceptable.

Reviewer 3 Report (New Reviewer)

Comments and Suggestions for Authors

This is a well-designed, hypothesis-driven study examining ketamine enantiomers and metabolites on ER stress and inflammation in human microglial cells. The focus on CHOP and GRP78 as molecular mediators of antidepressant action is timely and relevant. The experimental design is generally sound, and results are clearly presented with adequate controls. Yet the manuscript would benefit from additional mechanistic details, more descriptive figure legends, and some reorganization for easier reading. The detailed comments below are for further improvement.

Major Comments

1. The topic is current, but novelty is not as apparent in the abstract and discussion. For instance, in what ways is it pushing beyond existing ketamine-ER stress research?

2. The hypothesis needs to be stated explicitly in the Introduction. It is currently inferred but not explicitly stated.

3. While GRP78 protein levels were reduced, HSPA5 gene expression was not significantly affected—this discrepancy needs to be explained.

4. The manuscript would be strengthened by including experiments to target upstream regulators (e.g., PERK, IRE1) to confirm activation of specific UPR branches.

5. Report the n-value for each experiment. While ANOVA is correctly used, replicate and sample size clarity are lacking.

6. The manuscript assumes direct proportionality between mRNA and protein levels; this is not always the case. Some discussion of possible post-transcriptional regulatory processes would be useful.

7. The results are solely based on HMC3 cells. Acknowledging this limitation and suggesting future confirmation by primary microglia or in vivo models would add importance to the findings.

8. The authors mention that ELISA findings were conflicting, making them rely on Western blot for CHOP. Explain briefly why ELISA failed and whether or not this could be related to assay sensitivity or epitope issues.

Minor Comments

9. Use consistent terminology (e.g., "S-ketamine" vs. "S-ket"). Clarify time points and units directly in figure captions.

10. The Discussion section repeats results too much. Try shortening repetitive material to make room for additional interpretation.

11. The abstract must distinctly articulate the parameters of the research and specify the cellular model employed (HMC3 microglia). Furthermore, it is advisable to rephrase the final sentence to enhance clarity and effectiveness.

12. Inconsistent use of abbreviations like "UPR," "TM," and "HNKs." Define all on first use and be consistent.

13. Minor formatting mistakes (e.g., spaces before/after units, unnecessary semicolons) are found throughout. Professional proofreading is suggested.

14. IL-8 reduction data are not thoroughly investigated. Why is the effect less pronounced than that of IL-6? Is this cytokine more resistant to UPR modulation?

15. The conclusion reads more like a summary. Try to make it read more like it is emphasizing wider implications and what's next (e.g., translational significance, need for in vivo validation).

Comments on the Quality of English Language

The manuscript is generally composed of simple and clear English, but there are parts that need to be revised in terms of grammar, sentence structure, and clarity.

Author Response

Answer to Review 3

We thank the reviewer for their thoughtful and constructive feedback. We have carefully considered each comment and provide our detailed responses below.

Major Comments

  1. The topic is current, but novelty is not as apparent in the abstract and discussion. For instance, in what ways is it pushing beyond existing ketamine-ER stress research?

To clarify the novelty of our study, we have revised the Abstract and Discussion to explicitly highlight how our work advances current knowledge in the field with your suggestion.

Previous studies, such as Abelaira et al. (2017), have shown that ketamine modulates ER stress-related markers in the brain, particularly through the mTOR signaling pathway. However, our study introduces several novel aspects:

  • Cellular model focus on microglia:
    Most prior research was performed in vivo or focused on neurons. We specifically investigated microglial cells, which are central to neuroinflammation. This highlights the immune–stress axis in depression and the microglia-specific effects of ketamine and its metabolites.
  • Direct comparison of (S)- and (R)-ketamine and their metabolites (2S,6S-HNK and 2R,6R-HNK):
    To our knowledge, this is the first study to compare the enantiomers and their major metabolites side-by-side in terms of their effects on ER stress and pro-inflammatory cytokines in glial cells.
  • Integrated ER stress–inflammation readout:
    We provide an integrated view of the UPR pathway markers (CHOP, GRP78) alongside cytokine response (IL-6, IL-8) in the same experimental conditions.

These elements distinguish our study from existing work and support its translational value, especially in the context of treatment-resistant depression where both ER stress and neuroinflammation play a critical role.

  1. The hypothesis needs to be stated explicitly in the Introduction. It is currently inferred but not explicitly stated.

As suggested, we have now explicitly stated the study hypothesis at the end of the Introduction section.

  1. While GRP78 protein levels were reduced, HSPA5 gene expression was not significantly affected—this discrepancy needs to be explained.

The discrepancy between HSPA5 mRNA and GRP78 protein levels may reflect post-transcriptional regulatory mechanisms, including mRNA stability, translational efficiency, or protein degradation. It is well established that mRNA and protein levels do not always correlate directly, particularly for chaperone proteins like GRP78, whose expression is tightly controlled at multiple levels. We have now added a brief explanation of this point to the Discussion section.

  1. The manuscript would be strengthened by including experiments to target upstream regulators (e.g., PERK, IRE1) to confirm activation of specific UPR branches.

In both our previous and current studies, we aimed to investigate the involvement of all three main UPR branches in the mechanism of ketamine action. However, in both astrocytes and microglial cells, the most consistent and pronounced changes were observed for CHOP and GRP78. We agree that future studies would benefit from a more detailed characterization of individual UPR branches, including functional assays or analysis of phosphorylation status, to confirm the activation of upstream regulators.

  1. Report the n-value for each experiment. While ANOVA is correctly used, replicate and sample size clarity are lacking.

We have supplemented the manuscript with the previously missing n-values for all experiments.

  1. The manuscript assumes direct proportionality between mRNA and protein levels; this is not always the case. Some discussion of possible post-transcriptional regulatory processes would be useful.

We agree with the Reviewer’s comment. Although we did not observe significant changes in HSPA5 mRNA expression, we chose to assess GRP78 protein levels as well, precisely because we recognized that post-transcriptional regulation could play a role. As suggested, we have now added a short discussion addressing this issue and highlighting potential mechanisms such as altered translation efficiency or protein turnover that may explain the observed discrepancy.

  1. The results are solely based on HMC3 cells. Acknowledging this limitation and suggesting future confirmation by primary microglia or in vivo models would add importance to the findings.

According to the Reviewer's suggestion, we have now included a paragraph in the Discussion section addressing the limitations of our study.

  1. The authors mention that ELISA findings were conflicting, making them rely on Western blot for CHOP. Explain briefly why ELISA failed and whether or not this could be related to assay sensitivity or epitope issues.

In our initial experiments, the ELISA test for CHOP gave variable and unreliable results between replicates. This was likely due to the low sensitivity and overall quality of the kit used, which may not have been suitable for detecting small amounts of nuclear proteins like CHOP. The variability in results may also be due to the antibodies in the ELISA kit not binding effectively to the CHOP protein. Therefore, we decided to confirm CHOP expression by Western blot, which provided more consistent and reliable results in our model. In line with best practices, confirmation of protein levels using complementary methods such as ELISA and Western blot is often necessary when initial detection proves inconclusive. We have added a brief explanation to the Results section.

Minor Comments

  1. Use consistent terminology (e.g., "S-ketamine" vs. "S-ket"). Clarify time points and units directly in figure captions.

We have carefully reviewed the manuscript and made the following corrections:

- The full term “S-ketamine” is consistently used throughout the main text of the manuscript.

- Due to space constraints in figure panels, the abbreviations were retained in the figures themselves, but it has been clearly defined and explained in each figure caption.

- Additionally, time points and units have been included directly in all figure captions to meet formatting expectations and ensure clarity for the reader.

  1. The Discussion section repeats results too much. Try shortening repetitive material to make room for additional interpretation.

The Discussion section has been revised in accordance with suggestions.

  1. The abstract must distinctly articulate the parameters of the research and specify the cellular model employed (HMC3 microglia). Furthermore, it is advisable to rephrase the final sentence to enhance clarity and effectiveness.

The abstract has been revised according to your suggestion.

  1. Inconsistent use of abbreviations like "UPR," "TM," and "HNKs." Define all on first use and be consistent.

This has been corrected.

  1. Minor formatting mistakes (e.g., spaces before/after units, unnecessary semicolons) are found throughout. Professional proofreading is suggested.

This has been corrected.

  1. IL-8 reduction data are not thoroughly investigated. Why is the effect less pronounced than that of IL-6? Is this cytokine more resistant to UPR modulation?

We agree with the Reviewer that the reduction in IL-8 levels was less pronounced than that of IL-6, which is an important observation. This may reflect differences in their regulatory mechanisms. IL-6 expression is more closely associated with ER stress and CHOP-mediated signaling, which were affected in our model. In contrast, IL-8 is predominantly regulated by alternative pathways, such as NF-κB or AP-1, which may be less sensitive to UPR modulation (Hayden et al., 2008; Lawrence et al. 2002). These distinctions highlight the complexity of cytokine regulation and suggest that IL-8 may be more resistant to modulation by ketamine and its metabolites under the conditions tested. Interestingly, a similar observation was reported by SokoÅ‚owska et al. (2024), who demonstrated that ER stress (induced by different concentrations of tunicamycin) modulated IL-6 and IL-8 release from astrocytes and microglia, showing distinct cytokine responses compared to LPS stimulation.

  1. The conclusion reads more like a summary. Try to make it read more like it is emphasizing wider implications and what's next (e.g., translational significance, need for in vivo validation).

The Conclusions section has been revised to emphasize the broader implications of our findings, including translational relevance and the potential of ketamine metabolites as novel therapeutic candidates.

Comments on the Quality of English Language

The manuscript is generally composed of simple and clear English, but there are parts that need to be revised in terms of grammar, sentence structure, and clarity.

In response to the reviewers' comments regarding language quality, the manuscript was carefully revised for grammar, clarity, and sentence structure, and was subsequently checked by a native English speaker. A signed confirmation is attached.

Reviewer 4 Report (New Reviewer)

Comments and Suggestions for Authors

The manuscript presents an in vitro investigation into the effects of ketamine enantiomers and their metabolites on ER stress and inflammatory signaling in human microglial cells. The topic is timely and relevant, especially given the need for novel antidepressant strategies targeting treatment-resistant depression. However, the manuscript requires major revisions before it can be considered for publication.

The English throughout the manuscript is grammatically weak, inconsistent, and at times difficult to follow. Many sentences are long, repetitive, and awkwardly structured. The authors frequently restate the same ideas in different sections. A thorough language edit by a fluent English speaker or professional editing service is recommended.

The statistical analysis is generally appropriate but underreported. Although the authors mention the use of ANOVA and Tukey’s post hoc tests, they do not consistently provide full statistical details such as degrees of freedom, exact p-values, or whether assumptions of normality and variance homogeneity were tested. The use of fold change thresholds for gene expression (>1.5) is not well justified, and no rationale is provided for this cutoff. Sample sizes per group are also not clearly reported. 

The figures need to be revised. In particular, the data shown from Western blot and ELISA analyses are currently presented only as bar graphs with mean ± SEM, which is insufficient. Protein expression results, especially from Western blots, are subject to considerable variability and should be displayed with individual data points superimposed on the bars. This applies to all bar graphs summarizing protein quantification. 

The scientific content is potentially interesting, but the interpretation of the results is often overstated. The claim that CHOP and GRP78 represent novel therapeutic targets is not supported by the current in vitro data alone and should be softened. Similarly, the conclusion that these compounds have therapeutic potential in depression requires more cautious framing, especially given the lack of in vivo data and the use of a single immortalized microglial cell line. There is little discussion of the limitations of the model system, including the relevance of HMC3 cells to primary microglia and in vivo conditions. Additionally, the authors mention inconsistencies in CHOP ELISA results but do not transparently explain the limitations of switching to Western blotting or the number of replicates used.

Many paragraphs in the Introduction and Discussion repeat information or include speculative commentary that is not directly supported by the data. Statements about ketamine’s interaction with the mTOR pathway or its clinical efficacy are sometimes used without proper context or citation. The Discussion should be revised to clearly distinguish between the observed experimental findings and their potential clinical implications.

Author Response

Answer to Review 4

We are grateful to the Reviewer for the thoughtful and detailed comments, which have helped us improve the clarity and quality of the manuscript. We have addressed each point carefully in our revised version and responses below.

Comments and Suggestions for Authors

The manuscript presents an in vitro investigation into the effects of ketamine enantiomers and their metabolites on ER stress and inflammatory signaling in human microglial cells. The topic is timely and relevant, especially given the need for novel antidepressant strategies targeting treatment-resistant depression. However, the manuscript requires major revisions before it can be considered for publication.

The English throughout the manuscript is grammatically weak, inconsistent, and at times difficult to follow. Many sentences are long, repetitive, and awkwardly structured. The authors frequently restate the same ideas in different sections. A thorough language edit by a fluent English speaker or professional editing service is recommended.

The manuscript has been checked by a native English speaker. A signed confirmation is attached.

The statistical analysis is generally appropriate but underreported. Although the authors mention the use of ANOVA and Tukey’s post hoc tests, they do not consistently provide full statistical details such as degrees of freedom, exact p-values, or whether assumptions of normality and variance homogeneity were tested. The use of fold change thresholds for gene expression (>1.5) is not well justified, and no rationale is provided for this cutoff. Sample sizes per group are also not clearly reported. 

We have revised the manuscript to include full statistical details. Sample sizes (n) for each group are now explicitly reported in all figure captions.

The fold-change threshold of >1.5 used in the gene expression heatmap was not intended as a statistical cut-off, but rather as a visual tool to illustrate relative differences in expression (i.e., Δ vs. tunicamycin alone). This graphical representation was introduced to help readers intuitively distinguish the extent of downregulation. A similar approach was used in our previous publication involving astrocytes, where—in contrast to the present findings—ketamine and its metabolites increased UPR gene expression. This divergence highlights the importance of cell-type-specific responses in in vitro models.

The figures need to be revised. In particular, the data shown from Western blot and ELISA analyses are currently presented only as bar graphs with mean ± SEM, which is insufficient. Protein expression results, especially from Western blots, are subject to considerable variability and should be displayed with individual data points superimposed on the bars. This applies to all bar graphs summarizing protein quantification. 

In response, we have revised the Western blot figure (Fig. 4) to include individual data points superimposed on the bars, as suggested. This change enhances the presentation of variability in protein expression.

For the ELISA-based figures (Fig. 3, 5 and 6), we carefully evaluated the addition of individual data points. However, due to the number of groups and time points, overlaying individual values substantially reduced the clarity and legibility of the graphs. To maintain readability while ensuring transparency, we have retained the bar graphs with mean ± SEM and clearly indicated sample sizes (n) in the figure captions. We believe this strikes a balance between data precision and visual clarity.

The scientific content is potentially interesting, but the interpretation of the results is often overstated. The claim that CHOP and GRP78 represent novel therapeutic targets is not supported by the current in vitro data alone and should be softened. Similarly, the conclusion that these compounds have therapeutic potential in depression requires more cautious framing, especially given the lack of in vivo data and the use of a single immortalized microglial cell line. There is little discussion of the limitations of the model system, including the relevance of HMC3 cells to primary microglia and in vivo conditions.

The manuscript has been revised in accordance with the suggestions. We have reorganized the manuscript to enhance its structure and readability. We have also added a dedicated paragraph in the Discussion section to address the limitations of our study.

Additionally, the authors mention inconsistencies in CHOP ELISA results but do not transparently explain the limitations of switching to Western blotting or the number of replicates used.

In our initial experiments, the ELISA test for CHOP gave variable and unreliable results between replicates. This was likely due to the low sensitivity and overall quality of the kit used, which may not have been suitable for detecting small amounts of nuclear proteins like CHOP. The variability in results may also be due to the antibodies in the ELISA kit not binding effectively to the CHOP protein. Therefore, we decided to confirm CHOP expression by Western blot, which provided more consistent and reliable results in our model. In line with best practices, confirmation of protein levels using complementary methods such as ELISA and Western blot is often necessary when initial detection proves inconclusive. We have added a brief explanation to the Results section.

Many paragraphs in the Introduction and Discussion repeat information or include speculative commentary that is not directly supported by the data.

The Introduction and Discussion sections have been carefully revised. We ensured that all interpretations are now more clearly grounded in our data or supported by cited literature.

Statements about ketamine’s interaction with the mTOR pathway or its clinical efficacy are sometimes used without proper context or citation.

It has been corrected.

The Discussion should be revised to clearly distinguish between the observed experimental findings and their potential clinical implications.

The Discussion section has been revised in response to the reviewer’s comments.

Reviewer 5 Report (New Reviewer)

Comments and Suggestions for Authors

Comments

The manuscript with entitled “Modulation of ER stress and inflammation by (S)-ketamine, 2(R)-ketamine, and their metabolites in human microglial cells: Insights into novel targets for depression therapy”. It is interesting.

  1. This manuscript discusses the effect of modulation of ER stress and inflammation by (S)-ketamine, 2(R)-ketamine, therapeutic potential against nerve diseases and potential implications for health.

  1. The manuscript is original. The article provides materials for research in this field.

  1. In the introduction, the author should introduce the structure of (S)-ketamine and 2(R)-ketamine.

  1. What are the characteristic signs of the model of ER stress and inflammation of cell? How does the author determine the success of the cell model? The author should have clear indicators of successful model establishment. What is base of the dose of (S)-ketamine and 2(R)-ketamine used in this study? Recommendation: The author to provide supporting materials and additional clarification in the article.

.

  1. In the conclusions, what is the relationship between depression and ER stress and inflammation? The author should focus on discussing?.

  1. The references are appropriate, and it meets the requirements of the journal.

Author Response

Answer to Reviewer 5

We appreciate the reviewer’s insightful comments and suggestions to strengthen our manuscript. We have addressed each point in detail below:

In the introduction, the author should introduce the structure of (S)-ketamine and 2(R)-ketamine.

It has been done – the introduction now includes the structures of (S)-ketamine and (R)-ketamine, as well as their metabolites (Figure 1).

What are the characteristic signs of the model of ER stress and inflammation of cell? How does the author determine the success of the cell model? The author should have clear indicators of successful model establishment. What is base of the dose of (S)-ketamine and 2(R)-ketamine used in this study? Recommendation: The author to provide supporting materials and additional clarification in the article.

We agree that providing clear markers for model validation is essential.

To induce ER stress, we used tunicamycin, a well-established ER stress inducer, as a positive control. Likewise, lipopolysaccharide (LPS), a widely used pro-inflammatory agent, served as a positive control for the inflammation model. These controls were used to validate our model, and all tested compounds (S-ketamine, R-ketamine, and their metabolites) were evaluated in comparison to these conditions to assess their potential to modulate ER stress and inflammation.

To validate the success of these models:

  • For ER stress, we monitored the upregulation of key UPR pathway genes, including DDIT3 (CHOP) and HSPA5 (GRP78), following tunicamycin treatment (0.5 µg/mL). As shown in Figures 2 and 3, DDIT3 expression increased more than 12-fold, and HSPA5 more than 10-fold, indicating robust ER stress induction.
  • For inflammation, we assessed IL-6 and IL-8 secretion. LPS treatment (10 ng/mL) significantly increased both cytokines at 24 h and 48 h (Figure 5), confirming activation of an inflammatory response in microglial cells.

Regarding the dose selection of (S)-ketamine, (R)-ketamine, and their metabolites:

  • The concentration of 10 µM was selected based on preliminary MTT cytotoxicity tests, which confirmed it to be non-toxic (Supplementary Figure 1).
  • This dose also aligns with published literature, where clinical administration of ketamine (e.g., 0.5 mg/kg IV) results in plasma concentrations of ~1–4 µM. Since in vitro models often require higher concentrations to account for differences in uptake and metabolism, 10 µM is considered pharmacologically relevant (e.g. Zanos et al., 2018).

We have expanded the Methods and Results sections of the manuscript accordingly and added clarifying details to support these points (marked in red in the manuscript).

In the conclusions, what is the relationship between depression and ER stress and inflammation? The author should focus on discussing?.

 We thank the reviewer for pointing out this important aspect.

There is growing evidence that depression, particularly treatment-resistant depression (TRD), is associated with persistent neuroinflammation and endoplasmic reticulum (ER) stress, both of which contribute to impaired neuronal function and altered cellular homeostasis. When ER stress becomes prolonged or excessive, it activates the unfolded protein response (UPR), which can shift from adaptive to pro-apoptotic and pro-inflammatory signaling, involving proteins such as CHOP and GRP78. These stress pathways are known to interact with inflammatory responses mediated by cytokines like IL-6 and IL-8, which are often elevated in patients with depression.

In our study, we examined how ketamine enantiomers and their metabolites affect markers of ER stress and inflammation in microglial cells. The observed reduction in CHOP, GRP78, IL-6, and IL-8 levels supports the hypothesis that these pathways are modulated by ketamine-based compounds, which may underlie part of their antidepressant activity.

Importantly, the relationship between ER stress, inflammation, and depression has been previously described by Abelaira et al. (2017), who demonstrated that ketamine's antidepressant effects could involve modulation of ER stress markers through the mTOR signaling pathway. These findings provide a broader molecular context for our experimental results.

To emphasize this conceptual framework, we have:

  • Included relevant explanations in the Introduction, Discussion, and Conclusion sections,
  • And added a graphical abstract to visually summarize the interplay between ER stress, inflammation, and depressive pathophysiology, along with the potential therapeutic modulation by ketamine and its metabolites.

We hope this revised version provides a clearer integration of the mechanistic rationale and enhances the translational relevance of our findings.

Round 2

Reviewer 3 Report (New Reviewer)

Comments and Suggestions for Authors

No further comments

Reviewer 4 Report (New Reviewer)

Comments and Suggestions for Authors

I recommend acceptance in its current form.

Reviewer 5 Report (New Reviewer)

Comments and Suggestions for Authors

None

This manuscript is a resubmission of an earlier submission. The following is a list of the peer review reports and author responses from that submission.

Round 1

Reviewer 1 Report

Comments and Suggestions for Authors

Dear Marta Jóźwiak-BÄ™benista with colleagues,

Your study "Modulation of ER stress and inflammation by (S)-ketamine, 2 (R)-ketamine, and their metabolites in human microglial cells: 3 Insights into novel targets for depression therapy" is very interesting and novel. I do not have any issues/concerns/suggestions - well done!

Authors compared two isoforms of ketamine and their metabolites on ER stress pathways and inflammatory processes related to depression. R-ketamine is understudied and more research are needed in this direction. Hence, the main goal of the current manuscript is novel and worth to explore. 

This study attracts attention to potential of R-ketamine isoform. Effects of both isoforms were comparable, suggesting that R-ketamine is potent and potentially could elicit antidepressant effects. However, authors did not specify the source of S-ketamine and R-ketamine which is my major concern. Authors should add more information into the revised manuscript, what methods they used to separate S- vs R- ketamine isoforms, how they validated it. 

In addition to above mentioned missing information about S-, R-ketamine, also the statistical analysis is weak , authors need to report a main effect of drug treatment in a classical way , presenting F (df) ; p level) before describing individual comparisons.

Conclusions are logical and based on obtained results.

Author Response

Reviewer # 1

We sincerely thank the Reviewer for the very positive and encouraging feedback on our manuscript. We are pleased that the novelty of our study and the relevance of (R)-ketamine research were recognized.

Reviewer Comment:
“Authors did not specify the source of S-ketamine and R-ketamine which is my major concern. Authors should add more information into the revised manuscript, what methods they used to separate S- vs R- ketamine isoforms, how they validated it.”

Author Response:
We have now added detailed information to the Materials and Methods section regarding the origin and purity of the ketamine enantiomers used in our study.

Reviewer Comment:
“The statistical analysis is weak. Authors need to report a main effect of drug treatment in a classical way, presenting F(df); p-level before describing individual comparisons.”

Author Response:
We have revised the statistical reporting in the Results section to include the appropriate ANOVA results (F-values, degrees of freedom, and p-values) for the main effects of drug treatment, prior to describing post-hoc comparisons. This change improves clarity and adheres to standard statistical reporting conventions.

We hope these revisions adequately address the Reviewer’s comments, and once again, we are grateful for the constructive and supportive feedback.

Reviewer 2 Report

Comments and Suggestions for Authors

The article entitled “Modulation of ER stress and inflammation by (S)-ketamine,  (R)-ketamine, and their metabolites in human microglial cells:  Insights into novel targets for depression therapy” provides interesting results on the effects of ketamine isomers and their metabolites on UPR components during ER stress and neuroinflammation on microglial cells. However there are some issues that have to be addressed in order to make it suitable for publication.

  • The concentrations of drugs and substances inducing ER stress and inflammation were selected on the base of authors previous results and literature data: however, auth prev results were obtained on a different cell line, so it’s necessary to add tritations, at least as supplemental figures; also, authors must explain how the concentration chosen for ketamines is correlated to the ones used for therapeutic use in patients.
  • In table 1 add a bracket to link 10uM to all the ketamines (or repeat it for every one).
  • In fig 1 legends explaining what bars represent are missing (grey one represent TM alone, I guess).
  • In fig 1 B the illustration of the differences in gene expression is very confusing.
  • Why the protein levels of GRP78 has been evaluated only with ELISA and CHOP only with WB? In order to confirm results about both proteins, It’s necessary to evaluate the expression of the two proteins with both ELISA and WB.
  • Why the authors, to evaluate the effects of ketamines on LPS stilmulated microglia, chose IL-8, a neutrophil chemoattractant? It’s much more appropriate to analize IL-1b expression, the first and more powerful pro-infiammatory IL relased following LPS stimulation.
  • Original Images for Blots/Gels are useless without detailed captions
Comments on the Quality of English Language

It's necessary to improve many sentences, e.g.:

  • These processes, regulated by  the unfolded protein response (UPR), are increasingly recognized as promising TARGETS for INNOVATIVE therapeutic PROTOCOLS.
  • Elevated concentrations OF WHAT?, activity of  immune cells and increased levels of pro-inflammatory cytokines have been observed...
  • It is well established that circulating immune cells and their activation products WHAT ARE "ACTIVATION PRODUCTS" ?
  • inflammatory cytokines attack numerous molecular cell structures in activated glial 67 cells, including e.g. ER. CKs "ATTACK" INTRACELLULAR ORGANELLES?
  • The precise mechanism OF ACTION? of ketamine remains unclear

and so on.

Author Response

Reviewer # 2

We would like to sincerely thank the Reviewer for the careful and constructive evaluation of our manuscript. Below, we provide point-by-point responses to all comments raised by the Reviewer.

Reviewer Comment:

The concentrations of drugs and substances inducing ER stress and inflammation were selected on the base of authors previous results and literature data: however, auth prev results were obtained on a different cell line, so it’s necessary to add tritations, at least as supplemental figures; also, authors must explain how the concentration chosen for ketamines is correlated to the ones used for therapeutic use in patients.

Author Response:

The concentration of 10 µM used for S-ketamine, R-ketamine, and their metabolites in our in vitro experiments was selected based on both our previous work (e.g. Jóźwiak-BÄ™benisat et al., 2022) and published data on ketamine pharmacokinetics. While direct extrapolation between in vitro and in vivo concentrations is not straightforward—due to the absence of metabolism, protein binding, and blood–brain barrier effects in cell culture systems—clinical studies have reported that intravenous administration of ketamine (e.g., 0.5 mg/kg over 40 minutes) leads to plasma levels of approximately 0.3–1 µg/mL (corresponding to ~1–4 µM) (Zanos et al., 2018; Domino, 2010). Taking into account local accumulation and tissue exposure, the use of 10 µM in vitro is considered pharmacologically relevant for cellular mechanisms in vitro. This clarification has been added to the manuscript.

Furthermore, in response to the reviewer’s request, we have included additional MTT assay results performed on microglial cells. We tested a full range of concentrations (0.1, 1, 10, and 100 µM) for all compounds studied (S-ketamine, R-ketamine, and their metabolites). The results confirmed that 10 µM is non-toxic and falls within the safe concentration range for further evaluation. These data have been added as Supplementary Figure 1 in the revised version of the manuscript.

Reviewer Comment:

In table 1 add a bracket to link 10uM to all the ketamines (or repeat it for every one).

Author Response: It has been corrected.

Reviewer Comment:

In fig 1 legends explaining what bars represent are missing (grey one represent TM alone, I guess).

Author Response: It has been corrected.

Reviewer Comment:

In fig 1 B the illustration of the differences in gene expression is very confusing.

Author Response: It has been corrected.

Reviewer Comment:

Why the protein levels of GRP78 has been evaluated only with ELISA and CHOP only with WB? In order to confirm results about both proteins, It’s necessary to evaluate the expression of the two proteins with both ELISA and WB.

Author Response:

We thank the reviewer for the valuable comment regarding the methodological consistency in the evaluation of GRP78 and CHOP protein levels. GRP78 was assessed using a commercially available ELISA kit (EIAab Science, Wuhan, China), which allowed for sensitive and quantitative analysis.

Initially, we also attempted to measure CHOP protein levels using ELISA; however, this approach yielded inconsistent and technically unreliable results. In light of this, and considering the critical importance of validating potential negative findings with an alternative method, we employed Western blotting to ensure specific detection and confirm the observed expression pattern of CHOP.

We agree that evaluating both proteins using both ELISA and Western blotting would be ideal. Future studies will include cross-validation of GRP78 and CHOP with both methods. We prioritized using the most technically robust and reproducible approach available for each target. This methodological rationale has been added to the revised manuscript (Results section).

Reviewer Comment:

Why the authors, to evaluate the effects of ketamines on LPS stilmulated microglia, chose IL-8, a neutrophil chemoattractant? It’s much more appropriate to analize IL-1b expression, the first and more powerful pro-infiammatory IL relased following LPS stimulation.

Author Response:

While IL-1β is indeed a well-established and potent pro-inflammatory cytokine released following LPS stimulation, our choice to evaluate IL-8 was intentional. IL-8 (CXCL8) is a chemokine involved in neuroinflammatory processes and has been shown to be produced by activated microglia, contributing to leukocyte recruitment, neuroimmune interactions, and glia–neuron communication. Importantly, IL-8 has been identified as a potential biomarker in depression and treatment-resistant depression (Tsai, 2021).

Importantly, we chose IL-8 precisely because it is less commonly studied in the context of microglial activation and ketamine treatment. By doing so, we aimed to provide additional, novel insight into the broader cytokine profile modulated under inflammatory conditions.

Our findings on IL-8 complement the IL-6 results and strengthen the interpretation of ketamine's immunomodulatory potential. Nevertheless, we agree that IL-1β is an important cytokine and plan to include it in future studies to expand the cytokine panel.

Reviewer Comment:

Original Images for Blots/Gels are useless without detailed captions

Author Response: The original Western blot images have now been re-uploaded with detailed captions.

Reviewer Comment:

Comments on the Quality of English Language

It's necessary to improve many sentences, e.g.:

  • These processes, regulated by  the unfolded protein response (UPR), are increasingly recognized as promising TARGETS for INNOVATIVE therapeutic PROTOCOLS.
  • Elevated concentrations OF WHAT?, activity of  immune cells and increased levels of pro-inflammatory cytokines have been observed...
  • It is well established that circulating immune cells and their activation products WHAT ARE "ACTIVATION PRODUCTS" ?
  • inflammatory cytokines attack numerous molecular cell structures in activated glial 67 cells, including e.g. ER. CKs "ATTACK" INTRACELLULAR ORGANELLES?
  • The precise mechanism OF ACTION? of ketamine remains unclear

and so on.

Author Response: The sentences noted by the reviewer, along with the overall manuscript, have been revised.

Reviewer 3 Report

Comments and Suggestions for Authors

This submitted manuscript focuses on the anti-inflammatory roles of ketamine and its metabolites in human microglial cells. Overall, the results are straightforward and the whole structure of the presentation is clear.  However, this manuscript has several major concerns.

1) the anti-inflammatory role of ketamine is not novelty. Many previous publications have demonstrated that.

2) The whole project only employed one human microglial cell line.  All data were from in vitro studies.  NO in vivo data was involved in this manuscript.  It is not a qualified paper for Cells. 

3) Fig. 5 is confusing to the reviewer.  There are no effects of ketamine and the metabolites on proinflammatory factor under ER stress condition.  How to explain the difference between Fig. 4 and Fig. 5? 

Author Response

Reviewer # 3

We are grateful to the Reviewer for the thorough evaluation and valuable feedback on our manuscript. We have addressed each comment carefully, and our responses are provided below.

Reviewer Comment:

  • the anti-inflammatory role of ketamine is not novelty. Many previous publications have demonstrated that.

Author Response:

We fully agree that the anti-inflammatory properties of ketamine have been previously described in various models. However, the goal of our study was to explore the connection between inflammation and UPR pathway as a novel aspect of antidepressant action. To our knowledge, the first study to suggest a potential interaction between ketamine, mTOR signaling, and ER stress was conducted by Abelaira et al. (2017), who used racemic ketamine in a rodent model. However, their work was limited to in vivo conditions and did not assess enantiomer-specific effects or use human cell systems. Our study introduces several novel elements:

  1. We used human microglial cells (HMC3) to examine cell-type-specific responses in a model with higher translational relevance to human neuroinflammation than commonly used rodent systems.
  2. We compared the effects of (R)-ketamine, (S)-ketamine, and their hydroxynorketamine metabolites, which is rarely addressed in previous studies, especially in the context of ER stress.
  3. We investigated the dual action of these compounds on both ER stress and inflammatory cytokine release, highlighting a mechanistic link between these two pathways that are both implicated in the pathophysiology of depression.
  4. We showed that the anti-inflammatory effects of ketamine and its metabolites are context-dependent, being significant under LPS stimulation but not under tunicamycin-induced ER stress, indicating distinct modes of action in different cellular stress conditions.

To better illustrate the novelty of our study, we include a brief comparison table showing how our experimental model and focus differ from previous key publications in the field.

Study

Cell/Model Type

Substance Tested

ER Stress or related pathways

Human Microglia

Enantiomer or Metabolite Comparison

Abelaira et al., 2017

In vivo (rats, PFC infusion)

Racemic ketamine

CHOP, IRE1-α, PERK (via mTOR)

No

No

Ho et al., 2021

HMC3 (human microglia)

(2R,6R)-HNK, (2S,6S)-HNK

No (only IFN/STAT3 pathway)

Yes

Yes

Rafało-Ulińska et al., 2022

CUMS model in mice (PFC, hippocampus)

(R)- and (S)-ketamine

mTOR, ERK, BDNF/TrkB

No

No

Wang et al., 2022

Mouse (post-op depression model)

(S)-ketamine

No

No

No

Zou et al., 2021

Mouse microglia (BV-2)

Racemic ketamine

No

No

No

Our study

Human microglial cells (HMC3)

(S)-ketamine, (R)-ketamine, and metabolites

Yes

Yes

Yes

Taken together, while the anti-inflammatory action of ketamine has been reported, our study introduces a novel idea on how enantiomers and metabolites act on the inflammation–ER stress axis in human microglia. This supports their potential as modulators of neuroinflammation and ER-related cellular stress in depression.

Reviewer Comment:

  • The whole project only employed one human microglial cell line.  All data were from in vitro studies.  NO in vivo data was involved in this manuscript.  It is not a qualified paper for Cells. 

Author Response:

We thank the Reviewer for this observation and fully acknowledge that in vivo studies provide critical insights into complex biological systems. However, we respectfully disagree that the absence of in vivo data disqualifies our manuscript from consideration in Cells.

Our use of an in vitro model was a deliberate and necessary step to explore the specific cellular mechanisms by which (S)-ketamine, (R)-ketamine, and their metabolites modulate inflammation and ER stress responses in human microglial cells. In vitro systems offer unique advantages, such as:

  • The ability to study cell-type-specific effects in isolation, free from systemic variables present in in vivo models.
  • The possibility to dissect molecular mechanisms (e.g., ER stress-related pathways) with greater precision.
  • The use of human-derived cells, which improves translational relevance compared to many rodent-based in vivo studies.

Importantly, our findings extend previous work in human astrocytes (Jóźwiak-BÄ™benista et al., 2022), highlighting cell-type-specific effects of ketamine in the CNS. These mechanistic insights are not only valuable in their own right but also serve as a critical foundation for designing and justifying future in vivo studies. We are currently preparing a grant proposal to pursue in vivo validation of these findings in relevant models of depression.

Reviewer Comment:

3) Fig. 5 is confusing to the reviewer.  There are no effects of ketamine and the metabolites on proinflammatory factor under ER stress condition.  How to explain the difference between Fig. 4 and Fig. 5? 

Author Response:

The key difference between Figure 4 and Figure 5 lies in the type of stimulus used to activate microglial cells. Figure 4 presents results from an inflammatory model induced by lipopolysaccharide (LPS), a potent activator of Toll-like receptor 4 (TLR4), which triggers a rapid and robust proinflammatory response. In this context, both ketamine isomers and their metabolites significantly reduced IL-6 levels and, to a lesser extent, IL-8 secretion.

In contrast, Figure 5 shows results obtained under endoplasmic reticulum (ER) stress conditions, induced by tunicamycin (TM), which activates the unfolded protein response (UPR) pathway. This stress model engages different signaling mechanisms and a slower, more adaptive cellular response. Although ketamine and its metabolites modulated UPR-related gene and protein expression (e.g., CHOP, GRP78), their effect on IL-6 and IL-8 release was not statistically significant in this condition.

These findings suggest that the anti-inflammatory effects of ketamine may depend on the nature of the cellular stressor. While LPS induces a classical inflammatory cascade where ketamine shows strong effects, ER stress may require different mechanisms or earlier intervention points. We have clarified this issue in the revised version of the Discussion. Future in vivo studies, may help to further clarify these differences and better reflect the physiological relevance of our findings.
